# Bovine pain scale: A novel tool for pain assessment in cattle undergoing surgery in the hospital setting

Rubia Mitalli Tomacheuski[1,2]*, Cassandra Klostermann[3], Diane Frank[3], Marilda Onghero Taffarel[4], Renata Haddad Pinho[4,5], Beatriz Paglerani Monteiro[3], Pedro Henrique Esteves Trindade[1,6], André Desrochers[3], Sylvain Nichols[3], Karina Gleerup[7], Stelio Pacca Loureiro Luna[1,8], Paulo Vinicius Steagall[1,3,9,10]

1 Department of Anaesthesiology, Medical School (FMB) of São Paulo State University (UNESP), Botucatu, São Paulo, Brazil, 2 Translational Research in Pain, Department of Clinical Sciences, College of Veterinary Medicine, North Carolina State University (NCSU), Raleigh, North Carolina, United States of America, 3 Department of Clinical Sciences, Faculty of Veterinary Medicine, Université de Montréal, Saint-Hyacinthe, Québec, Canada, 4 Department of Veterinary Medicine, Maringa State University (UEM), Umuarama, Paraná, Brazil, 5 Faculty of Veterinary Medicine, University of Calgary, Calgary, Alberta, Canada, 6 Department of Large Animal Clinical Sciences, College of Veterinary Medicine, Michigan State University (MSU), East Lansing, Michigan, United States of America, 7 Department of Clinical Sciences, University of Copenhagen, Taastrup, Denmark, 8 Department of Veterinary Surgery and Animal Reproduction, School of Veterinary Medicine and Animal Science, São Paulo State University (UNESP), Botucatu, São Paulo, Brazil, 9 Department of Veterinary Clinical Sciences, Jockey Club College of Veterinary Medicine and Life Sciences, City University of Hong Kong, Hong Kong S.A.R., China, 10 Centre for Companion Animal Health and Welfare, City University of Hong Kong, Hong Kong S.A.R., China

* rubiamitalli11@gmail.com

## Abstract

Pain negatively impacts animal welfare and it is still neglected in ruminants. This original study aimed to develop and validate the Bovine Pain Scale (BPS) for acute pain assessment in hospitalized cattle undergoing surgery. This was a blinded, randomized, prospective clinical study. Thirty-six animals were included in the study. The Pain Group (n = 25) included patients admitted to a veterinary teaching hospital requiring any type of soft tissue or orthopedic surgery. Videos were recorded before, 2–6 hours after surgery, 1 hour after the administration of analgesia and 24 hours after surgery. The Control Group (n = 11) included healthy animals that were video recorded twice within a 24-48h interval. The BPS was developed using content validity. A total of 118 videos of 6 minutes were randomized and analyzed by four raters who were unaware of groups, time-points and procedures in two phases with a five-week interval. Statistical analysis was performed using R software. Intra and inter-rater reliability (intra-class correlation coefficient) was very good (0.83–0.94) and ranged from good to very good, respectively (0.65–0.81). The correlation between the BPS and the Visual Analog Scale (VAS) was strong (rho = 0.77, p < 0.0001) confirming criterion validity. Item-total correlation was acceptable for 3 of 9 items (0.33–0.43) and internal consistency was below the acceptable value (0.6). The scale

**Data availability statement:** Data is available in the supporting information.

**Funding:** This study was funded in part by the Coordenação de Aperfeiçoamento de Pessoal de Nível Superior - Brasil (CAPES- PrInt) - process number 88887.466964/2019-00; and Brazil's National Council for Scientific and Technological Development (CNPq) - process number 140349/2018-9; which provided a doctorate scholarship for the first author. The founders had no role in data collection and analysis, decision to publish, or preparation of the manuscript.

**Competing interests:** The authors have declared that no competing interests exist.

was responsive to pain but not the administration of analgesia. It was specific for five items, but no items showed sensitivity. The area under the curve of 0.90 demonstrated high discriminatory capacity. According to the receiver operating characteristic curve, the cut-off point for rescue analgesia was ≥ 5 of 18. The BPS is reliable and reproducible, showed content and criterion validity, and may be used in veterinary hospitals for assessing post-operative pain in cattle to guide decision-making towards rescue analgesia. Future studies should refine the instrument to guarantee construct validity and sensitivity.

## Introduction

Animal welfare is a social and ethical concern [1,2], and it has become a priority to improve husbandry practices [3,4]. Pain negatively impacts animal welfare; it causes fear and suffering, and decreases productivity (e.g., weight gain, milk production, reproductivity) [5–7]. Veterinarians have a responsibility to alleviate pain in animals. Yet, farm animals are less frequently administered analgesics when compared with companion animals [8]. A study with Canadian beef cattle producers showed that in 90% of castration, 85% of dehorning procedures and 46% of cases of dystocia, animals did not receive any analgesia [9]. Historically, beef cattle receive less analgesics than dairy cattle [10]. Although there has been an increase in the use of analgesics and pain recognition by veterinarians working with cattle [11,12], pain management is still neglected in this species [7].

Pain scales are used for pain assessment. They need to undergo rigorous validation to ensure they are measuring what they are supposed to measure (e.g., pain) and consequently, can be used in clinical practice [13]. Ideally, a pain scale should be applicable to animals with different painful conditions (i.e., medical or surgical pain) as well as in different environments (i.e., hospitals, farms). Previous work includes the Cow Pain Scale (CPS) [14], which was developed in dairy cattle with clinical pain and the UNESP-Botucatu cattle pain scale (UCAPS) [15] that was developed in beef cattle undergoing orchiectomy in extensive farm production. Although the UCAPS has been validated in the hospital setting, animals underwent a single procedure (orchiectomy) and were assessed in groups and in paddocks [16]. Nevertheless, both scales are limited to the populations and scenarios in which they were validated and cannot be generalized to a large population of animals with different types of pain. Currently, the literature is lacking a pain assessment instrument that can be used in a variety of surgical procedures using different anesthetic and analgesic dosage regimens in hospitalized cattle.

### Objective & hypothesis

This study aimed to develop and validate the Bovine Pain Scale (BPS) for pain assessment after surgery in cattle in the hospital setting. The hypothesis was that the BPS is a reliable and valid instrument, that discriminates painful and non-painful individuals and responsive to the administration of analgesia.

## Materials and methods

The development and validation of the BPS was based on a clinical prospective and observational study performed at the Farm Animal Hospital of the Centre Hospitalier Universitaire Vétérinaire (CHUV) of the Faculté de médecine vétérinaire (FMV), Université de Montréal, a referral hospital that receives over 1000 farm animals every year. Its modern hospital is the clinical center of reference of Quebec, Canada, offering cutting-edge diagnosis and treatment of farm animals with advanced equipment and surgery as well as training programs. The study was approved by the institutional animal care and use committee of the Université de Montréal (protocol 20-Rech-2065) and followed the recommendations of the COSMIN [17,18].

### Animals

Forty-three animals of any sex or breed and older than 6 months were included in the study (recruitment period from July 21st 2020 to January 28th 2021). The Pain Group consisted of patients admitted to the CHUV requiring any type of soft tissue or orthopaedic surgery. Exclusion criteria included systemic disease (e.g., septicaemia or animals considered critically ill requiring intensive or immediate treatment) or severe lameness. Animals that were administered analgesics and/or sedatives up to 24h before admission or those with medical pain were not included. The anaesthetic and analgesic protocols and treatment plan for each animal were determined by the attending clinician on a case-by-case basis (Supplementary material Table S1). Upon arrival and admission, each animal was assessed for eligibility. A signed written informed consent form was obtained from the owner or farm manager before inclusion. The Control Group consisted of animals from the teaching colony of the FMV that were deemed healthy based on history, physical examination and regular blood work.

### Bovine pain scale

The preliminary version of the BPS consisted of 65 behaviours divided into 9 items in English and developed by investigators RMT, BPM, SPLL and PVS using their personal experience, and the pain-related behaviours described in the UCAPS [15] and the Cow Pain Scale [14]. Each item has scores 0 – normal and pain free behavior; scores 1 and 2 pain related behaviors. Content validity was assessed by eight veterinarians with experience in bovine practice (DA, CK, MOT and five collaborators). Additionally, pain assessment was performed in seven animals by RMT to test the feasibility of the BPS. After the calculation of content validity index (Supplementary Material), the definitive version of the Bovine Pain Scale used in this study is presented in Table 1.

### Real-time pain assessment and video recording

Real-time pain assessment by one female veterinarian (RMT) included four and two different time-points in the Pain and Control Group, respectively, while animals were in their single stalls. Time-points were as follows: P1, preoperatively, immediately prior to administration of sedatives; P2, 2–6 hours after the end of surgery; P3, 1 hour after the administration of analgesic intervention if required; P4, 24 hours after surgery in the Pain Group. In the Control group, time-points involved C1 (according to the availability of the animal when not involved in teaching) and C2 (24–48 hours after C1). At P3, rescue analgesia was administered if pain scores were ≥ 5 of 10 according to the UCAPS with the agreement of the attending clinician. Rescue analgesia generally involved the administration of an opioid (e.g., butorphanol) and/or a non-steroidal anti-inflammatory drug (NSAID).

Video recordings were performed simultaneously to real-time pain assessment using two cameras. One camera (GoPro Hero 5, GoPro, San Mateo, CA, USA) was placed at a higher location to capture a top view of the animal while the other (Canon PowerShot G16, 5x optical zoom lens 6.1–30.5mm, Oita, Japan) was placed in front of the stall for a frontal view of the animal. The duration of video recordings was six minutes and the cameras were set up 15 minutes before the first evaluation time-point. For the subsequent time-points, a 5-minute acclimation was performed. The observer was

**Table 1. The Bovine Pain Scale consists of nine items with inclusion of three descriptive levels. Score 0 means normal behaviours (pain free), score 1 and 2 behaviours related to pain.**

| ITEM | VARIABLE | VIDEO LINK |
|---|---|---|
| **Appetite** | (0) Normorexia and/or rumination | https://drive.google.com/file/d/10GQ-Fw2rJblzCh_sY9RAlsoiuzMFpXmNu/view?usp=sharing |
| | (1) Hyporexia | https://drive.google.com/file/d/1kFOCPeVLirBAM_yDxF5tdTPJYrk1tBx6/view?usp=sharing |
| | (2) Anorexia | https://drive.google.com/file/d/1Uw4w-VDn9pBanYKhlPJO5PZvoezdP9tSG/view?usp=sharing |
| **Posture when standing** | Arching the back (except when standing up or urinating) | https://drive.google.com/file/d/1C9Mxv-vRbOA9_4tqqgkoJO1myDcb8wKff/view?usp=sharing |
| | Hind limbs extended caudally (observe from the side) | https://drive.google.com/file/d/1IPppDDM-WPBCl92mqMj295Y0mfA17natl/view?usp=sharing |
| | Top of the head below the line of spinal column (if not eating) | https://drive.google.com/file/d/189knbkOoN-Q2WIrTWVtl9yo41OMEHp8tr/view?usp=sharing |
| | (0) All of the above-described behaviours are absent | |
| | (1) Presence of 1 of the above-described behaviours | |
| | (2) Presence of 2 or more of the above-described behaviours | |
| **Posture when lying down** | Ventral recumbency with full or partial extension of one or both hind limbs | https://drive.google.com/file/d/16vEf4ajMnrxpn6GpjbJAuDvcXMw0UYga/view?usp=sharing |
| | Head on/close to the ground | https://drive.google.com/file/d/1ZnNmlDGxyEj6vs5UvrSIrGa7zQ6tpiwg/view?usp=sharing |
| | Extending the neck and body forward when in ventral recumbency | https://drive.google.com/file/d/1g6c0NXEzQdr5ME9HFXMj8uZjbfrtdz-9/view?usp=sharing |
| | (0) All of the above behaviours are absent | |
| | (1) Presence of 1 of the above-described behaviours | |
| | (2) Presence of 2 or more of the above-described behaviours | |
| **Miscellaneous behaviours 1** | Groaning | https://drive.google.com/file/d/1v-Orqb-Zz5kNGLdsNYclb7kQQqvQn41TW/view?usp=sharing |
| | Attention towards the painful area | https://drive.google.com/file/d/1prWK-OGlxJGF6hahCxFcX3HYptpCRF3W6/view?usp=sharing |
| | Licking the surgical wound | https://drive.google.com/file/d/1WHz71jJ_ksnvH2p_Oz5x32LB0yrO2TLy/view?usp=sharing |
| | (0) All of the above behaviours are absent | |
| | (1) Presence of 1 of the above-described behaviours | |
| | (2) Presence of 2 or more of the above-described behaviours | |

*(Continued)*

**Table 1.** (Continued)

| ITEM | VARIABLE | VIDEO LINK |
|---|---|---|
| **Miscellaneous behaviours 2** | Lambs' ears, ears rotated back and the pinna facing down | https://drive.google.com/file/d/1p3ns-ZCN3X-AbIUbWqdwDLruCmtBXi12S/view?usp=sharing |
| | Tense expression/strained appearance, furrows above the eyes and puckers above the nostrils | https://drive.google.com/file/d/1lIZF7Iq_J4zjj9kGRliH_AzQBUQQ4tXy/view?usp=sharing |
| | Wagging the tail abruptly and repeatedly | https://drive.google.com/file/d/1DTIsyEzO4giaWrjoDrvFcUUhH4XaFNFX/view?usp=sharing |
| | (0) All of the above behaviours are absent | |
| | (1) Presence of 1 of the above-described behaviours | |
| | (2) Presence of 2 or more of the above-described behaviours | |
| **Limb movement/ condition** | Lifting one foot of the ground | https://drive.google.com/file/d/1Icsjdw1vS246h0cOhkFA1VrPFYyoq5P-/view?usp=sharing |
| | Kicking/foot stamping | https://drive.google.com/file/d/16ewg-peMK5E5gLeuFTThuU09mENNUFKuE/view?usp=sharing |
| | Restlessness (pacing) | https://drive.google.com/file/d/1JiiB0_5w-1p7JIEcEU8MZDE9-e.g.,5NboJ1/view?usp=sharing |
| | Weight shifting | https://drive.google.com/file/d/1WbsrS-PA-W7D8vgVP7MnvVD3S6ZA31N_B/view?usp=sharing |
| | (0) All of the above-described behaviours are absent | |
| | (1) Presence of 1 of the above-described behaviours | |
| | (2) Presence of 2 or more of the above-described behaviours | |
| **Interactive behaviour with the environment** | (0) Active and attentive to environmental stimuli. When near other animals, can interact with and/or accompany the group | https://drive.google.com/file/d/1VHuY-JmUrKfwbxkLOB2fd4599VU0GYJWz/view?usp=sharing |
| | (1) Apathetic, interacting little when stimulated. When near other animals might remain close to them | https://drive.google.com/file/d/1tKL-cV49KTEbkZOyey_TAn_uwZV0TJmLh/view?usp=sharing |
| | (2) Apathetic; not reacting to environmental stimuli. When near other animals may be isolated or not accompany them | https://drive.google.com/file/d/1L9uPJIv4j4XQQHRC9jd6Q8e_6lNyzpu0/view?usp=sharing |
| **Response to approach** *(if the animal is lying down or standing still stimulate it with clapping hands)* | (0) Animal's head up, ears forward, or may interrupt briefly ongoing activity (grooming, ruminating, etc.) | https://drive.google.com/file/d/1aLPrHo5GujMGynqRkIwK0JEbsntu7cMc/view?usp=sharing |
| | (1) Animal's ears not forward, orients by moving head in the direction of the observer | https://drive.google.com/file/d/1tPYQ8pe6_xNEvA7BcoAar9LVP1Z3ut4_/view?usp=sharing |
| | (2) Animal's ears back, head low, does not orient head toward the observer clapping hands (no head movement) | https://drive.google.com/file/d/1HCtFEVwlRbLM-QjImP_NCkWsfKrzatzG/view?usp=sharing |
| **Activity and locomotion** *(if the animal is lying down or standing still stimulate it with clapping hands)* | (0) Moving normally. Walking with no obviously abnormal gait, or relaxed in ventral recumbency position (resting quietly), or standing still easily or eating or ruminating | https://drive.google.com/file/d/1WClix5-3iOiLPjdfBGpd0VqUUfkt9e9y/view?usp=sharing |
| | (1) Walking with restriction, hunched back when moving or short steps. May be agitated (constant changes in weight-bearing) or laying restlessness (difficult to find a comfortable position) | https://drive.google.com/file/d/1HhdXy-TPuIZJQCGzkX7b-jwIdMm8-c_Z/view?usp=sharing |
| | (2) Reluctant to stand up, standing up with difficulty or not walking | https://drive.google.com/file/d/1pzla6huHZ8wesiAURDmSs5nHrsNiFYjK/view?usp=sharing |

 

always present during acclimation and video recording. Animals were left undisturbed for the first 3 minutes of recordings. Thereafter, the observer approached, clapped her hands three times and then offered food (hay, silage, or dried feed) while recordings continued until the 6-minute mark. Pain scoring was then completed using the UCAPS [15], the visual analog scale (VAS; 0–100 mm, where 0 = no pain and 100 = the worst pain imaginable) and the BPS proposed herein.

## Pain assessment using video recordings

Video recordings were randomized for pain assessment via video analysis by five experienced veterinarians who were not aware of the group, time-point, procedure nor involved with recordings. However, surgical wounds may have been seen in some of these videos. A Training Manual (Supplementary Material) of the BPS was provided to raters prior to video assessment. It included a written description of each score for each behaviour/item and their respective link of a video-example (Google Drive). Videos were made available using a virtual platform (SurveyMonkey). Each video included the two simultaneous recordings from each camera and were presented as one video per page. The video-analysis was divided into Phase 1 and Phase 2 (Fig 1).

### Phase 1

This phase consisted of 10 videos (5–6 min/video = approximately 1h total) from seven animals that were not included in Phase 2. These videos were presented twice, 2 weeks apart, in a different randomized order (Part 1 and Part 2). The five raters (experienced veterinarians) received a questionnaire with the BPS including 12 randomized questions per video (Supplementary material). Nine questions were related to the BPS, one question presented the VAS for pain assessment as previously described and one question on whether they believed rescue analgesia was needed. They were asked to watch each video with sound and then answer each question. Videos could be watched as many times as needed. If any item of the BPS was not visible, raters could mark "not possible to score". Inter and intra-rater reliability analysis was carried out for Parts 1 and 2 using the intraclass correlation coefficient (ICC). Raters with ICC ≥ 0.80 for intra-reliability were invited to participate in Phase 2.

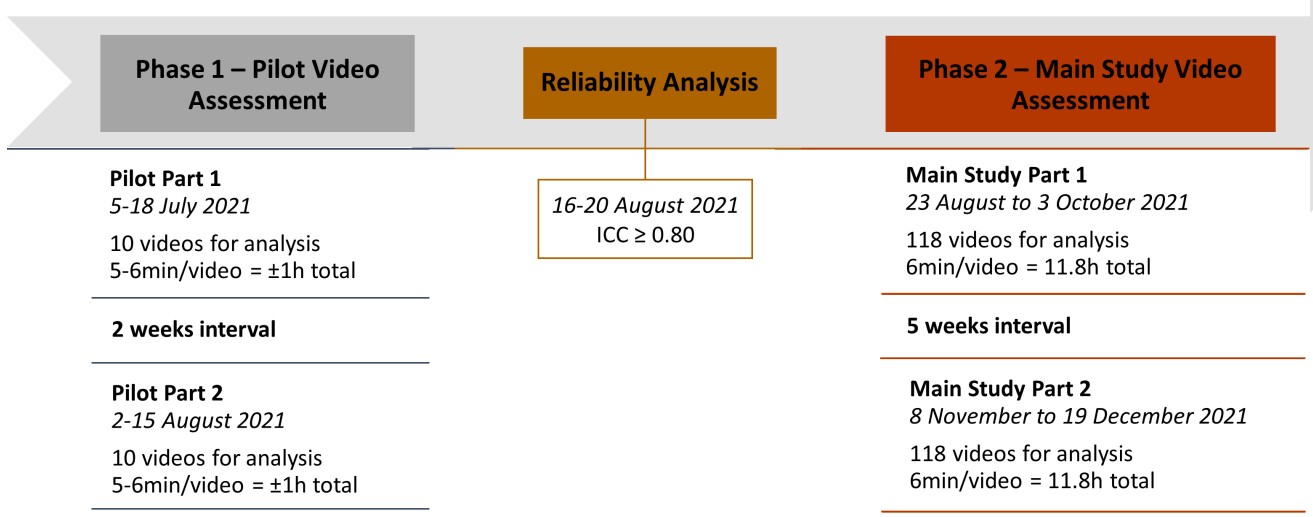

**Fig 1. Timeline of video analysis for the validation of the Bovine Pain Scale.**

## Phase 2

A total of 118 videos (6 min/video = 11.8h in total), from 36 animals (Pain Group - 25 animals; Control Group - 11 animals), were presented twice, 5 weeks apart, in a different randomization order (Phase 2; Part 1 and Part 2). Four raters (experienced veterinarians) were included and received a link every week with 20 videos for analysis during five weeks and 18 videos during the last week. Video assessment was performed as described for Phase 1. Raters were asked to not perform more than 1-hour of video assessment per day to avoid fatigue.

## Statistical analyses

Statistical analyses were performed by PHET using R software in the RStudio integrated development environment (RstudioTeam, 2016), using data from video analysis of Phases 1 (five raters; to test the reliability of raters to qualify for the next phase) and 2 (four raters; all time-points individually and grouped, and for both Pain and Control Groups). A $p < 0.05$ was considered for statistical significance. The Shapiro-Wilk test and the Gaussian distribution according to the quantile-quantile and histograms plots confirmed that data did not present normal distribution. Hence, nonparametric tests were carried out for analysis. Table 2 provides a detailed description of statistical analysis. A minimum sample size of 11 subjects, with 0.80 of power and an alpha of 0.05 was calculated, based on Spearman correlation of rho = 0.764 between the UCAPS and Cow Pain Scale (http://biomath.info/power/) [16].

# Results

Data collection was performed from July 2020 to January 2021. A total of 36 animals were included with 25 from the Pain Group (Table 3) and 11 from the Control Group. Seven animals were excluded from Phase 2 of study because their videos were used in Phase 1.

## Intra and inter-rater reliability

Intra-rater reliability was very good for all raters (> 0.80) for the BPS total score and VAS. It varied from very good to moderate for the BPS items (Table 4). Inter-rater reliability varied from good to very good for the BPS total score (≥ 0.65) and from good to moderate for VAS (Table 5). The items 'appetite' and 'posture when lying down' showed the best reproducibility (0.69–0.99). Other items varied from reasonable to good (0.22–0.83) (Table 5).

## Exploratory factor analysis

The exploratory factor analyses determined that the BPS is bidimensional (Table 6) [24]. All items, except the two miscellaneous behaviours, were representative in one of the dimensions.

## Criterion validity

The correlation between the BPS and VAS was strong with rho = 0.77 ($p < 0.0001$) (Fig 2), confirming concurrent criterion validity.

## Specificity and sensitivity

The BPS items 'appetite', 'posture when standing' and 'when lying down', 'miscellaneous behaviours 1', 'interactive behaviour and 'response to approach' were specific (Control group data). For the total score, specificity was close to 70%. Otherwise only 'miscellaneous behaviours' showed sensitivity because all other values, including the total score were lower than 70% (Table 7).

## Construct validity

   **Item-total correlation & Internal consistency.**  'Appetite', 'interactive behaviour with the environment' and 'activity and locomotion' were accepted according to the interpretation of correlation r with values between 0.3–0.7 (Table 8). The

**Table 2. Statistical methods used for validation of the Bovine Pain Scale (BPS) in cattle.**

| Type of analysis* | Description | Groups analysed | Statistical test |
|---|---|---|---|
| **Content validity** | A list of behaviours reported in the literature was scored by eight experts (experienced veterinarians). They analysed each item as either normal or pain-related behaviour using a content validity index with each item of the scale being highly relevant (4), quite-relevant or highly relevant, and needs rewording (3), somewhat relevant (2), or not relevant (1). Also rating into relevant (+1), do not know (0), or irrelevant (-1). | N/A | All the values of each subitem (-1, 0, or 1) were added and the total was divided by the number of experts. Items with a total score > 0.5 were included in the scale [19,20] |
| **Intra-rater reliability** | Repeatability - the level of agreement between each rater with themselves by comparing the two phases of assessment, using the scores of each item and the total sum of the BPS and the need for rescue analgesia. | Pain and Control groups | For the scores of the items of the BPS, the weighted kappa coefficient ($k_w$) was used; the disagreements were weighted according to their distance to the square of perfect agreement ("cohen.kappa{-psych}"). The 95% confidence interval (CI) $k_w$ based in 1,001 replications by the bootstrap method was estimated ("boot.ci{boot}"). For the sum of the BPS and VAS, the intraclass correlation coefficient (ICC; "icc{irr}") two-way random effects model, type consistency multiple raters/measurements and its 95% CI based in 1,001 replications by the bootstrap method ("boot.ci{boot}") was used. Interpretation of $k_w$ and ICC: very good 0.81–1.0; good 0.61–0.80; moderate 0.41–0.60; reasonable 0.21–0.4 and poor < 0.2 [21–23] |
| **Inter-rater reliability** | Reproducibility (agreement matrix) - the level of agreement between the four raters, using the scores for each item and the total sum of the scale. | Pain and Control groups | |
| **Multiple association** | The multiple association of the scale items with each other was analysed at all time-points grouped using principal component analysis, to define the number of dimensions determined by different variables that establish the scale extension. Exploratory factor analysis was used to compare one dimension, two uncorrelated dimensions and two correlated dimension models. | Pain and Control groups | Exploratory factor analysis (EFA) based on correlation matrix, a maximum likelihood method, and with oblimin rotation ("fa{-psych}"). Horn's Parallel Analysis was performed to determine the optimal number of factors to be retained in the EFA [24]. Loading value ≥ 0.50 or ≤ - 0.50 were considered for significant association. |
| **Criterion validity** | Concurrent criterion validity (comparison with another instrument)—the total score of BPS was correlated with the VAS. | Pain and Control groups | Spearman rank correlation coefficient (rs; "rcorr" function of the "Hmisc" package). Interpretation of the degree of correlation rs (p < 0.05): < 0.19 very weak,; 0.2–0.39 weak, 0.4–0.59 moderate, 0.6–0.79 strong and 0.8–1 very strong [21,25]. |
| **Specificity and Sensitivity** | Specificity: based on true negatives – number of cattle 'without pain' from the control group that should score '0' compared to the total number of cattle for each item (C1 + C2) and those that should score < 5 (please see the cut-off point description below) for the total sum of BPS<br>Sensitivity: based on true positives – number of cattle with scores 1 or 2 compared to the total number (pain group for each item (P2) and those showing total pain score ≥ 5 for the total score BPS. | Pain and Control groups | Specificity, sensitivity and its 95% CI were calculated according to the bootstrap method described below ("epi.tests{epiR}"). Interpretation: excellent 95–100%; good 85–94.9%; moderate 70–84.9%; not specific or not sensitive < 70% [26]. |
| **Item-total correlation** | The correlation of each item of the scale after excluding the evaluated item, were estimated to analyse homogeneity, the inflationary items, and the relevance of each item of the scale. Analysis was performed for all grouped time-points. | Pain and Control groups | Spearman rank correlation coefficient ($r_2$; "rcorr{Hmisc}"). Interpretation of item-total correlation r: suitable values 0.3–0.7 [21]. |

*(Continued)*

**Table 2.** (Continued)

| Type of analysis* | Description | Groups analysed | Statistical test |
|---|---|---|---|
| **Internal consistency** | The consistency (interrelation) of the scores of each item of the scale were calculated. The analysis was performed for all grouped time-points. | Pain and Control groups | McDonald's omega coefficient interpretation: 0.65–0.80, acceptable; > 0.80 strong reliability evidence [27] |
| **Responsiveness** | The scores should close to zero or low before pain stimulus, should increase after surgery and then decrease after analgesic intervention [21,28] | Pain group | For each item of the scale, the Friedman's test was used, followed by Dunn's multiple comparison post-hoc test [29] For the dichotomous variable, need for rescue analgesic, logistic regression analysis ("glm" function of the "stats" package) was applied using the post hoc Tukey test ("lsmeans" function of the "lsmeans" package). The normality of the model residuals ("residuals" function of the "stats" package) for the dependent variable (BPS) showed Gaussian distribution for groupings according to the quantile-quantile and histogram graphs ("qqnorm" and "histogram" functions of the "stats" and "lattice" packages, respectively), thus, mixed linear models ("lme" function of the "nlme" package) were applied. The residual distribution was not considered normal for other dependent variables (BPS separated by group) and, therefore, generalized mixed linear models ("glmer" function of the "lme4" package) were applied. For all models, time-points, raters, groups and phases were included as fixed effects and the individuals as random effect. Differences over time and intergroups were performed by the Bonferroni test as a post hoc test [30]. Modelling was used to conduct responsiveness. Then, a multilevel linear model (lme4::lmer) was applied to BPS and VAS, while a multilevel generalized linear model adjusted by Poisson distribution (lme4::glmer) was used to items of BPS, and a multilevel binomial logistic regression (lme4::glmer) was conducted to sub-items of BPS and the indication of analgesia. Timepoints, raters, phases, and times to watch each video were included as explanatory variables, while bovine was included as random effects of the model. Bonferroni was used for adjusting the multiple comparisons in the post-hoc test (lsmeans::lsmeans and multcomp::cld). Interpretation: differences in scores were expected to occur in the Pain group (P1≤P3≤P4<P2), and the scores should not change in the Control group. |
| **Construct validity** | 1. Internal relationships among items | Pain and Control groups | See internal consistency, item-total correlation, and principal component analysis. |
| | 2. Relationships to scores of other instruments | | See criterion validity. |
| **Optimum cut-off point** | Control group all time-points (C1 and C2) free of pain, and Pain group (pre and postoperative) were used to determine optimal cut-off point. The data relative to the indication of rescue analgesia according to the observer experience were used to determine the optimal cut-off point, that is, the minimum score suggestive of the need for analgesic rescue or intervention. | Pain and Control groups | Cut-off point was based on the Youden index [YI = (Sensitivity + Specificity) − 1], which determines the highest sensitivity and specificity value concurrently from the Receiver Operating Characteristic (ROC) curve ("roc{pROC}" and "ci.sp{pROC}"), providing a graphic image of the relation between the "true positives" (sensitivity) and the "true negatives" (specificity). The discriminatory capacity of the scale was determined by the area under the curve (AUC). AUC values above 0.90 represent high accuracy discriminatory capacity of the scale [31]. In addition, the 95% confidence interval (CI) was calculated from the Youden index by replicating the original ROC curve 1,001 times according to the bootstrap method ("ci.coords{pROC}" and "ci.auc{pROC}"). In addition, the diagnostic uncertainty zone was determined by two methods: 1) calculating the 95% confidence interval (CI) by replicating the original ROC curve 1000 times according to the bootstrap method. The lowest and highest values of these two methods among all raters were assumed to be the diagnostic uncertainty zone [31,32] |

Time-points: P1, preoperatively, immediately prior to administration of sedatives; P2, 2–6 hours after the end of surgery; P3, 1 hour after the administration of analgesic intervention if required; P4, 24 hours after surgery. C1, first video and assessment of animal of the control group; C2, the second video of the same animal of the control group, recorded 24–48 hours after C1.

**Table 3. Demographics and summarized history of animals included in the Pain Group.**

| Breed | Age Years | Body weight | Sex | BCS (1–9) | History | Type of stable | Surgery type |
|---|---|---|---|---|---|---|---|
| Holstein | 7 | 890 Kg | F | 5 | Difficulty milking front right teat | Tie stall | Theloscopy front right teat |
| Holstein | 5 | 695 kg | F | 4 | Difficulty milking front left teat. Suspecting of a varicose vein at the annular ring | Tie stall | Phlebectomy of varicose vein |
| Simental | 5 | 920 kg | M | 8 | Fractured left mandible. Difficulty eating | Box stall | Mandibular fracture |
| Holstein | 1 | 340 kg | F | 7 | Fractured left metacarpus | Box stall | Left metacarpal fracture |
| Holstein | 5 | 718 kg | F | 7 | Lumbar mass above spinal column | Box stall | Lumbar mass removal |
| Holstein | 2 | 608 kg | F | 7 | Difficulty milking hind left teat | Tie stall | Partial teat amputation |
| Holstein | 2 | 520 kg | F | 7 | Difficulty milking front right teat | Tie stall | Theloscopy front right teat |
| Blonde D'Aquitaine | 8 | 800 kg | F | 8 | Biopsy and mass removal; lactating with a calf | Box stall | Mass removal |
| Holstein | 7 | 650 kg | F | 6 | Metallic foreign body in the rumen | Box stall | Rumenotomy |
| Ayrshire | 5 | 610 kg | F | 7 | Mass in the left hock | Box stall | Left posterior hock hygroma |
| Holstein | 3 | 690 kg | F | 6 | Cranial peritonitis | Box stall | Rumenotomy |
| Holstein | 7 | 754 kg | F | 6 | Macerated veal | Tie stall | Caesarean section |
| Holstein | 5 | 607 Kg | F | 6 | Difficulty milking, laceration, and mammary gland infection hind right teat | Tie stall | Theloscopy hind right teat |
| Holstein | 5 | 782 kg | F | 6 | Left eye injury with exophthalmos and a mass | Box stall | Enucleation |
| Holstein | 4 | 774 kg | F | 7 | Cranial cruciate ligament rupture | Box stall | Cranial cruciate ligament repair |
| Holstein | 8 | 705 kg | F | 6 | Colic, pyloric mass, and atonic rumen | Box stall | Abomasopexy |
| Holstein | 3 | 555 kg | F | 7 | Left displacement of the abomasum, hypermotility of rumen | Box stall | Laparoscopy and omentopexy |
| Holstein | 6 | 764 kg | F | 6 | Ovarian mass | Tie stall | Laparoscopy and ovariectomy |
| Holstein | 4 | 680 kg | F | 6 | Left elbow abduction, evidence of cranial peritonitis | Tie stall | Rumenotomy |
| Holstein | 6 | 800 kg | F | 7 | Mass in the right eye | Box stall | Enucleation |
| Holstein | 6 | 825 kg | F | 7 | Colic | Box stall | Abomasopexy |
| Limousin | 1 | 340 kg | F | 8 | Lameness in the left hindlimb | Box stall | Stifle/ meniscus repair |
| Holstein | 5 | 698 kg | F | 6 | Caesarean section for educational purposes | Box stall | Caesarean |
| Holstein | 5 | 650 kg | F | 7 | Colic | Box stall | Rumenotomy |
| Holstein | 4 | 660 kg | F | 5 | Rumen impaction, hypermobile and muscle fasciculation | Box stall | Laparoscopy and abomasopexy |

https://doi.org/10.1371/journal.pone.0323710 May 23, 2025

**Table 4. Intra-rater reliability of the Bovine Pain Scale, unidimensional scales and indication for rescue analgesia in the perioperative period of cattle undergoing surgery (n = 25; Pain Group) and in healthy cows (n = 11; Control Group).**

| | Rater 1 | Rater 2 | Rater 3 | Rater 4 |
|---|---|---|---|---|
| **Variables** | **Weighted Kappa (Lower - Upper)** | **Weighted Kappa (Lower - Upper)** | **Weighted Kappa (Lower - Upper)** | **Weighted Kappa (Lower - Upper)** |
| Rescue Analgesia | 0.51 (0.35 - 0.65) | **0.80** (0.67 - 0.89) | 0.52 (0.31 - 0.69) | 0.50 (0.35 - 0.65) |
| Appetite | **0.94** (0.89 - 0.98) | **0.97** (0.92 - 0.99) | **0.96** (0.93 - 0.99) | **0.94** (0.88 - 0.99) |
| Posture when standing | 0.52 (0.33 - 0.65) | **0.79** (0.65 - 0.9) | **0.65** (0.51 - 0.76) | **0.68** (0.51 - 0.79) |
| Posture when lying down | **0.83** (0.73 - 0.90) | **0.96** (0.91 - 0.99) | **0.80** (0.69 - 0.89) | **0.85** (0.78 - 0.91) |
| Miscellaneous behaviours 1 | **0.77** (0.60 - 0.91) | **0.72** (0.54 - 0.86) | 0.59 (0.43 - 0.72) | **0.75** (0.58 - 0.89) |
| Miscellaneous behaviours 2 | 0.51 (0.36 - 0.65) | **0.83** (0.75 - 0.90) | 0.55 (0.4 - 0.69) | 0.57 (0.43 - 0.71) |
| Limb movement/condition | **0.70** (0.60 - 0.8) | **0.85** (0.79 - 0.91) | **0.78** (0.70 - 0.85) | **0.67** (0.53 - 0.79) |
| Interactive behaviour with the environment | 0.53 (0.33 - 0.69) | **0.77** (0.59 - 0.91) | **0.69** (0.59 - 0.77) | 0.44 (0.26 - 0.58) |
| Response to approach | **0.78** (0.66 - 0.88) | **0.88** (0.78 - 0.95) | **0.88** (0.80 - 0.94) | **0.74** (0.60 - 0.85) |
| Activity and locomotion | **0.71** (0.52 - 0.85) | **0.92** (0.85 - 0.97) | **0.70** (0.58 - 0.81) | 0.52 (0.34 - 0.68) |
| | **ICC (CI Lower - CI Upper)** | **ICC (CI Lower - CI Upper)** | **ICC (CI Lower - CI Upper)** | **ICC (CI Lower - CI Upper)** |
| **Total Score BPS** | **0.83** (0.76 - 0.88) | **0.94** (0.92 - 0.96) | **0.91** (0.87 - 0.94) | **0.89** (0.84 - 0.92) |
| **VAS** | **0.82** (0.74 - 0.87) | **0.95** (0.93 - 0.96) | **0.88** (0.83 - 0.92) | **0.81** (0.73 - 0.87) |

Notes: BPS, Bovine Pain Scale, scores 0–18; VAS, Visual analog scale, scores 0–100 mm, where 0 = no pain and 100 = the worst pain imaginable. Statistical tests: ICC, Intraclass correlation coefficient; CI, Confidence interval 95%. Interpretation of reliability: very good 0.81–1.0; good 0.61–0.80; moderate 0.41–0.60; reasonable 0.21–0.40; poor < 0.20 [21,22,33]. Bold values ≥ 0.61. ICC demonstrated as multiple measures formula for UCAPS and CPS, ICC model: alpha, two-way mixed; type: consistency, and absolute agreement for VAS. Rescue analgesia was indicated based on the raters' response to "Do you think it is necessary to provide rescue analgesia?": yes (1) or no (0).

McDonald's omega coefficient for the calculation of internal consistency was not acceptable for any item as all values were lower than 0.65 (Table 8). However, the fact that internal consistency of the items: 'appetite', 'posture when lying down', 'interactive behaviour' and 'activity and locomotion' reduced after their exclusion means that they somewhat contribute to the full scale.

## Responsiveness

The total score of the BPS was significantly higher in P2 than in P1 and P4, confirming responsiveness; however, responsiveness was not observed after the administration of analgesia (Table 9 and Fig 3). There was no responsiveness for VAS (Table 9).

The phase of the study (1 or 2) did not have any significant effect on the BPS total score or VAS (Table 9 and Supplementary Material Fig S1), whereas 'raters' and the number of times videos were watched affected both scales (Table 9 and Supplementary Material Fig S2). There were significant differences between time-points for some BPS items, except for the item 'posture when standing', 'miscellaneous behaviour 2', 'response to approach' and 'activity and locomotion'.

## Optimum cut-off point for rescue analgesia

The receiver operating characteristic (ROC) curve showed that scores of ≥ 5 of 18 discriminated painful vs non-painful individuals. The resampling confidence interval > 0.90 for the Youden index was between 4.5 and 5.5, therefore score 5 is within the diagnostic uncertainty zone, scores ≤ 4 are true negatives and ≥ 6 are true positives. The area under the curve (AUC) was 0.90 indicating a high discriminatory capacity (Fig 4).

**Table 5. Inter-rater reliability of the Bovine Pain Scale, unidimensional scales, and indication of rescue analgesia in the perioperative period of cattle undergoing surgery (n = 25; Pain Group) and in healthy cows (n = 11; Control Group).**

| Variables | Rater 1 *versus* Rater 2 Weight Kappa (Lower - Upper) | Rater 1 *versus* Rater 3 Weight Kappa (Lower - Upper) | Rater 1 *versus* Rater 4 Weight Kappa (Lower - Upper) | Rater 2 *versus* Rater 3 Weight Kappa (Lower - Upper) | Rater 2 *versus* Rater 4 Weight Kappa (Lower - Upper) | Rater 3 *versus* Rater 4 Weight Kappa (Lower - Upper) | All Raters *versus* All Raters Weight Kappa (Lower - Upper) |
|---|---|---|---|---|---|---|---|
| Rescue Analgesia | 0.24 (0.13 - 0.35) | 0.22 (0.14 - 0.29) | 0.16 (0.04 - 0.28) | 0.35 (0.22 - 0.47) | 0.37 (0.26 - 0.48) | 0.22 (0.12 - 0.33) | 0.33 (0.16 - 0.32) |
| Appetite | **0.90** (0.85 - 0.94) | **0.88** (0.83 - 0.91) | **0.94** (0.90 - 0.97) | **0.92** (0.87 - 0.95) | **0.88** (0.83 - 0.93) | **0.88** (0.84 - 0.92) | **0.92** (0.84 - 0.91) |
| Posture when standing | 0.39 (0.28 - 0.49) | 0.47 (0.35 - 0.58) | 0.57 (0.44 - 0.68) | 0.57 (0.46 - 0.67) | 0.39 (0.27 - 0.5) | 0.42 (0.31 - 0.52) | 0.52 (0.32 - 0.5) |
| Posture when lying down | **0.79** (0.71 - 0.86) | **0.69** (0.59 - 0.78) | **0.83** (0.76 - 0.88) | **0.67** (0.57 - 0.76) | **0.82** (0.75 - 0.87) | **0.72** (0.63 - 0.8) | **0.80** (0.68 - 0.81) |
| Miscellaneous behaviours 1 | **0.67** (0.53 - 0.79) | **0.62** (0.50 - 0.72) | 0.56 (0.42 - 0.68) | **0.65** (0.52 - 0.76) | 0.59 (0.45 - 0.72) | 0.53 (0.40 - 0.64) | **0.64** (0.51 - 0.69) |
| Miscellaneous behaviours 2 | 0.39 (0.26 - 0.5) | 0.34 (0.22 - 0.45) | 0.32 (0.19 - 0.44) | 0.42 (0.31 - 0.53) | 0.57 (0.48 - 0.65) | 0.40 (0.29 - 0.49) | 0.49 (0.35 - 0.52) |
| Limb movement/ condition | **0.63** (0.54 - 0.72) | 0.67 (0.59 - 0.74) | **0.63** (0.55 - 0.7) | 0.55 (0.46 - 0.63) | 0.50 (0.41 - 0.59) | **0.61** (0.52 - 0.7) | **0.70** (0.52 - 0.64) |
| Interactive behaviour with the environment | 0.42 (0.25 - 0.57) | 0.22 (0.15 - 0.29) | 0.38 (0.24 - 0.51) | 0.26 (0.19 - 0.34) | 0.43 (0.29 - 0.55) | 0.30 (0.21 - 0.38) | 0.38 (0.20 - 0.35) |
| Response to approach | 0.51 (0.41 - 0.61) | **0.75** (0.66 - 0.83) | 0.58 (0.48 - 0.68) | **0.63** (0.53 - 0.71) | 0.33 (0.24 - 0.43) | 0.57 (0.47 - 0.66) | **0.66** (0.46 - 0.61) |
| Activity and locomotion | **0.78** (0.67 - 0.86) | 0.41 (0.24 - 0.56) | 0.29 (0.09 - 0.47) | 0.38 (0.26 - 0.51) | 0.29 (0.12 - 0.46) | 0.31 (0.17 - 0.44) | 0.44 (0.26 - 0.48) |
| | ICC (CI Lower - CI Upper) | ICC (CI Lower - CI Upper) | ICC (CI Lower - CI Upper) | ICC (CI Lower - CI Upper) | ICC (CI Lower - CI Upper) | ICC (CI Lower - CI Upper) | ICC (CI Lower - CI Upper) |
| **Total Score BPS** | **0.70** (0.61 - 0.77) | **0.75** (0.68 - 0.81) | **0.73** (0.65 - 0.79) | **0.73** (0.65 - 0.79) | **0.65** (0.55 - 0.73) | **0.81** (0.74 - 0.84) | **0.84** (0.49 - 0.65) |
| **VAS** | 0.58 (0.46 - 0.68) | **0.65** (0.55 - 0.73) | 0.60 (0.49 - 0.69) | **0.76** (0.69 - 0.81) | **0.73** (0.65 - 0.79) | **0.72** (0.64 - 0.78) | **0.78** (0.54 - 0.68) |

Notes: BPS, Bovine Pain Scale, scores 0–18; VAS, Visual analogue scale, scores 0–100 mm, where 0 = no pain and 100 = the worst pain imaginable. Statistical tests: ICC, Intraclass correlation coefficient; CI, Confidence interval 95%. Interpretation of reliability: very good 0.81–1.0; good 0.61–0.80; moderate 0.41–0.60; reasonable 0.21–0.40; poor < 0.20 [21,22,33]. Bold values ≥ 0.61. ICC demonstrated as single measures formula for UCAPS and CPS, ICC model: alpha, two-way mixed; type: consistency, and absolute agreement for VAS. Rescue analgesia was indicated based on the rater's response to the following question before scoring the pain scales "Do you think it is necessary to provide rescue analgesia?" yes (1) or no (0).

## Discussion

The BPS is a reliable instrument to be used for pain assessment in cattle undergoing surgery in a hospital setting. The scale incorporated five behaviours of CPS, all behaviours from UCAPS and included five new behaviours. To the authors' knowledge, the scale is the first to be developed in a clinical environment in cattle submitted to different surgical procedures, where the animal is alone and inside a stall. Specificity was attained, the scale showed criterion validity and a cut-off score to guide the administration of analgesia was identified as reported by UCAPS [15]. Because sensitivity was not adequate and some attributes of construct validity were suboptimal, the instrument requires further refinement. This highlights how challenging pain assessment is in cattle, especially when they are kept in individual stalls (box or tie) during the perioperative period influenced by the hospital environment, where fear and anxiety may confound pain assessment. Furthermore, their preoperative pain condition, diversity of surgical procedures and anaesthetic protocols may have influenced postoperative pain assessment and our results. Additionally, pain-induced behaviours after soft

**Table 6. Loading values and eigenvalues of the Bovine Pain Scale items based on exploratory factor analysis.**

| Items | Dimension 1 | Dimension 2 | Dimension 3 |
|---|---|---|---|
| | Loading value | Loading value | Loading value |
| Appetite | 0.00 | -0.03 | **1.01** |
| Posture when standing | **0.77** | 0.24 | 0.17 |
| Posture when lying down | **-0.77** | 0.42 | 0.12 |
| Miscellaneous behaviours 1 | 0.00 | -0.13 | 0.27 |
| Miscellaneous behaviours 2 | 0.26 | 0.44 | 0.23 |
| Limb movement/condition | **0.68** | 0.11 | -0.13 |
| Interactive behaviour with the environment | -0.07 | **0.78** | -0.02 |
| Response to approach | 0.21 | **0.53** | -0.19 |
| Activity and locomotion | -0.07 | **0.53** | 0.28 |
| Eigenvalue | 2.37 | 1.63 | 0.82 |

The structure was determined considering items with a load value ≥0.50 or ≤−0.50 (in bold) [34].

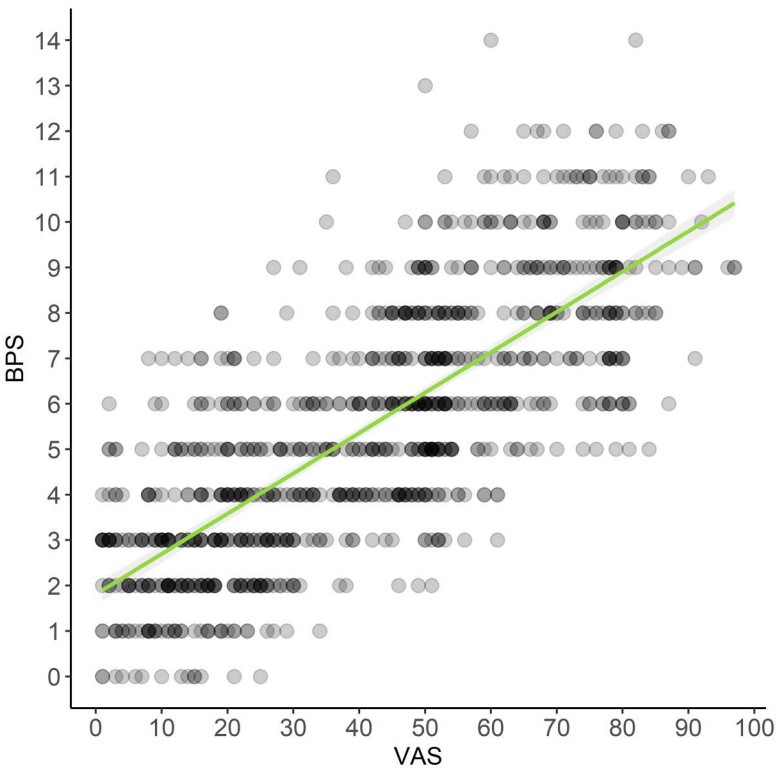

**Fig 2. Plot correlating the scores of the Bovine Pain Scale (BPS) and the Visual Analog Scale (VAS) of perioperative period of cattle undergoing surgery (n = 25; Pain Group) and in healthy cows (n = 11; Control Group).**

tissue and orthopaedic surgery may be expressed in different forms and intensity. For example, a single teat amputation may produce mild inflammation and can be treated with a non-steroidal anti-inflammatory drug alone. Caesarean section, rumenotomy and enucleation may require the administration of NSAIDs, local anaesthetics and opioids. As much as pain-induced behaviours may be presented differently and can vary with intensity and duration, the authors aimed to

**Table 7. Specificity and sensitivity of the Bovine Pain Scale.**

| Items | Specificity (%) (C1) | Specificity (%) (C1 + C2) | Sensitivity (%) (P2) |
|---|---|---|---|
| Appetite | **91** | **79** | 45 |
| Posture when standing | **80** | **73** | 45 |
| Posture when lying down | **82** | **71** | 38 |
| Miscellaneous behaviours 1 | **95** | **94** | 11 |
| Miscellaneous behaviours 2 | 21 | 15 | **82** |
| Limb movement/condition | 17 | 18 | 59 |
| Interactive behaviour | **93** | **85** | 46 |
| Response to approach | **81** | **79** | 51 |
| Activity and locomotion | 57 | 52 | 68 |
| **Total Score** | 69 | 57 | 65 |

Notes. Interpretation of specificity and sensitivity: excellent 95–100%; good 85–94.9%; moderate 70–84.9%; not specific or sensitive < 70%; bold values ≥ 70% [21].

**Table 8. Item-total correlation and internal consistency of the Bovine Pain Scale for all time-points.**

| Variables | Item-total (Spearman) | Internal consistency (McDonald's omega coefficient) |
|---|---|---|
| Full scale | – | 0.60 |
| | **Excluding each item below** | |
| **Items** | | |
| Appetite | **0.33** | 0.56 |
| Posture when standing | 0.19 | 0.64 |
| Posture when lying down | 0.08 | 0.56 |
| Miscellaneous behaviours 1 | 0.08 | 0.63 |
| Miscellaneous behaviours 2 | 0.27 | 0.61 |
| Limb movement/condition | -0.04 | 0.63 |
| Interactive behaviour with the environment | **0.41** | 0.52 |
| Response to approach | 0.24 | 0.62 |
| Activity and locomotion | **0.43** | 0.52 |

Interpretation of Spearman's rank correlation coefficient (r): 0.3–0.7 [21]. McDonald's omega coefficient interpretation: 0.65–0.80, acceptable; > 0.80 strong reliability evidence [27].

develop and validate a pain scoring instrument with wide applicability after surgery in hospitalized cattle. Future studies may target one type of procedure, or one source of pain and results may be more appropriate using, for example, either soft tissue or orthopaedic pain after surgery, like performed previously in beef [15] and dairy cattle [14].

The role of environment including the use of box or tie stalls with animals surrounded by unfamiliar sounds and smells is another challenge to validate a perioperative pain assessment tool in cattle. Additionally, the transportation and handling of an animal experiencing pain increases physiological stress [35]. The natural prey behaviour of cattle may add to the potential bias and challenge as pain may be masked and not recognized by veterinarians. All these challenges make pain assessment a difficult subject of study in this species.

The repeatability of BPS was very good and similar or superior to other pain scales developed for cattle [14,16,36,37]. The BPS assessment was well reproducible as inter-rater reliability for total scores was good or very good [21] and similar to UCAPS [15]. This might be a result of training for using the BPS before the study had begun. Reproducibility was better

**Table 9. Responsiveness of the Bovine Pain Scale (BPS), Visual Analog Scale (VAS) and rescue analgesia, between the four perioperative time-points, showed as median (first – third quartile) (n = 25).**

| Parameters | Time-points | | | | Effects on the model | | |
|---|---|---|---|---|---|---|---|
| | P1 | P2 | P3 | P4 | Rater | Phase | Times video was watched |
| Rescue Analgesia | 1 (0; 1) ab | 1 (0; 1) a | 1 (0; 1) ab | 1 (0; 1) b | < 0.0001 | 0.0856 | 0.0053 (β = 0.5576) |
| **Appetite** | 2 (0; 2) b | 0 (0; 2) a | 0 (0; 2) b | 0 (0; 2) b | 0.1075 | 0.3248 | 0.1167 (β = -0.1904) |
| **Posture when standing** | 0 (0; 1) | 0 (0; 1) | 0 (0; 0.5) | 0 (0; 1) | < 0.0001 | 0.5636 | 0.0002 (β = 0.4809) |
| Arching back | 0 (0; 0) c | 0 (0; 0) a | 0 (0; 0) ab | 0 (0; 0) bc | < 0.0001 | 0.3117 | 0.1349 (β = 0.7104) |
| Hind extended | 0 (0; 0) a | 0 (0; 0) ab | 0 (0; 0) ab | 0 (0; 0) b | 0.0745 | 0.7733 | 0.0205 (β = 0.8594) |
| Head below | 0 (0; 1) a | 0 (0; 1) a | 0 (0; 0) b | 0 (0; 1) a | < 0.0001 | 0.1362 | 0.0003 (β = 0.7969) |
| **Posture when lying down** | 0 (0; 0) c | 0 (0; 1.5) a | 0 (0; 1) bc | 0 (0; 1) ab | 0.1063 | 0.3590 | 0.1250 (β = -0.1974) |
| Ventral recumbency | 0 (0; 0) b | 0 (0; 1) a | 0 (0; 1) a | 0 (0; 1) a | 0.2235 | 0.2617 | 0.0315 (β = -0.5207) |
| Head ground | 0 (0; 0) b | 0 (0; 0.5) a | 0 (0; 0) c | 0 (0; 0) a | 0.0354 | 0.5565 | 0.3577 (β = -0.2566) |
| Extending neck recumbency | 0 (0; 0) b | 0 (0; 0) a | 0 (0; 0) b | 0 (0; 0) ab | < 0.0001 | 0.9731 | 0.3751 (β = -0.2993) |
| **Miscellaneous behaviours 1** | 0 (0; 1) a | 0 (0; 0) b | 0 (0; 1) b | 0 (0; 1) b | 0.0590 | 0.7120 | 0.2740 (β = 0.1822) |
| Groaning | 0 (0; 1) a | 0 (0; 0) c | 0 (0; 0) a | 0 (0; 0) b | 0.2073 | 0.8460 | 0.1113 (β = 0.5128) |
| Attention painful area | 0 (0; 0) b | 0 (0; 0) ab | 0 (0; 0) b | 0 (0; 0) a | <0.0001 | 0.3109 | 0.9568 (β = -0.0185) |
| Licking surgical wound | 0 (0; 0) | 0 (0; 0) | 0 (0; 0) | 0 (0; 0) | 0.2533 | 0.2487 | 0.9056 (β = -0.1381) |
| **Miscellaneous behaviours 2** | 2 (1; 2) | 1 (1; 2) | 1 (1; 2) | 1 (1; 2) | 0.0275 | 0.3083 | 0.5019 (β = 0.0522) |
| Ears rotated | 1 (0; 1) a | 1 (0; 1) ab | 1 (0; 1) bc | 0 (0; 1) c | < 0.0001 | 0.0058 | 0.2080 (β = 0.2744) |
| Tense expression | 0.5 (0; 1) a | 0 (0; 1) ab | 0 (0; 1) b | 0 (0; 1) ab | 0.0104 | 0.8609 | 0.8379 (β = -0.0408) |
| Wagging tail | 0 (0; 1) | 0 (0; 1) | 0 (0; 1) | 0 (0; 1) | 0.0219 | 0.6199 | 0.0634 (β = 0.3916) |
| **Limb movement/condition** | 0 (0; 1) bc | 1 (0; 2) a | 1 (0; 1) ab | 0 (0; 1) c | < 0.0001 | 0.8299 | 0.0116 (β = 0.2414) |
| Lifting one foot | 0 (0; 1) | 0 (0; 1) | 0 (0; 1) | 0 (0; 1) | < 0.0001 | 0.1197 | 0.0061 (β = 0.6014) |
| Kicking | 0 (0; 0) b | 0 (0; 0) a | 0 (0; 0) bc | 0 (0; 0) c | 0.0016 | 0.9150 | 0.9451 (β = 0.0228) |
| Restlessness | 0 (0; 0) | 0 (0; 0) | 0 (0; 0) | 0 (0; 0) | 0.0048 | 0.2516 | 0.0265 (β = 1.3577) |
| Weight shifting | 0 (0; 0) b | 0 (0; 1) ab | 0 (0; 1) a | 0 (0; 1) ab | < 0.0001 | 0.2148 | 0.0033 (β = 0.6493) |
| **Interactive behaviour with the environment** | 0 (0; 1) ab | 0 (0; 1) a | 0 (0; 1) a | 0 (0; 1) b | < 0.0001 | 0.5884 | 0.6516 (β = -0.0689) |

*(Continued)*

**Table 9.** (Continued)

| Parameters | Time-points | | | | Effects on the model | | |
|---|---|---|---|---|---|---|---|
| | **P1** | **P2** | **P3** | **P4** | **Rater** | **Phase** | **Times video was watched** |
| **Response to approach** | 0 (0; 2) | 1 (0; 2) | 0 (0; 2) | 0 (0; 1) | < 0.0001 | 0.3366 | 0.0328 (β = 0.2069) |
| **Activity and locomotion** | 1 (0; 1) | 1 (0; 1.5) | 1 (0; 1) | 1 (0; 1) | < 0.0001 | 0.9523 | 0.2724 (β = 0.1302) |
| **BPS total score** | **5 (4; 7) b** | **6 (3.5; 9) a** | **6 (4; 8) ab** | **5 (3; 7) b** | **0.0119** | **0.5529** | **0.0024 (β = 5.5312)** |
| VAS | 43 (25; 56) ab | 50 (23; 69) a | 46 (24; 59) ab | 37 (18; 56) b | 0.0023 | 0.2972 | 0.0018 (β = 6.6266) |

Time-points: P1, preoperatively, immediately prior to administration of sedatives; P2, 2–6 hours after the end of surgery; P3, 1 hour after the administration of analgesic intervention if required; P4, 24 hours after surgery; were assessed. Interpretation: a > b > c. BPS, Bovine Pain Scale; VAS, Visual analogue scale; Rescue analgesia was indicated based on the rater's response to the following question before scoring the pain scales "Do you think it is necessary to provide rescue analgesia?" yes (1) or no (0).

than VAS similarly to reported in sheep [30], showing the possible advantage of using a pain scale that includes several components of pain [38].

The exploratory factor analysis showed that the BPS is a bidimensional scale. This is the first bidimensional pain scale other than the Unesp-Botucatu multidimensional feline pain assessment scale [23]. All other pain behaviour scales developed and validated for cattle and other species were mathematically unidimensional [14,15]. This is an important measurement property of health instruments [39,40] as it identifies the number of domains of a scoring instrument [34,41]. The BPS includes various expressions of pain including physiological (appetite), sensory or motor (posture, limb movement/condition, activity) and emotional (interactive behaviour with the environment, response to approach, activity and locomotion). Therefore, as reported for CPS [14] and UCAPS [15], BPS is biologically multidimensional.

The criterion validity was confirmed by the strong correlation between BPS and VAS like previous studies in cattle [14,15,42]. Criterion validity is based on assessing the correlation between the new instruments *versus* a 'gold standard' [21]. The CPS and UCAPS may be classified as 'gold standards' instruments because they fit the COSMIN criteria used for validation of health metric instruments. For this reason, they were not compared with BPS [25,28,39,41], because the latter incorporated a combination of 100% of UCAPS and 37.5% of the behaviours of CPS. Comparisons with these items would lead to the overinflation of results.

The construct validity is the ability of a scale to measure what is supposed to be measured; pain in this case [21]. Sensitivity was inadequate; only three items presented adequate item-total correlation and internal consistency was below acceptable showing that the correlation between items and their interrelation was not appropriate. Although UCAPS had a better internal consistency and item-total correlation [14,15], it was tested only after orchiectomy using a standard analgesic and anaesthetic protocol. The current study was performed in a clinical setting including animals with different levels of pain and undergoing several types of surgeries, treatments and anaesthetic protocols resulting in heterogeneous behaviours, which may have affected internal consistency. Future studies should test the BPS in a homogenous population (animals, treatments, surgical procedures and anesthetic protocols) in an attempt to improve the internal consistency. The results of the specificity of BPS were suboptimal and inferior to the UCAPS and CPS [14,15], unless only the first time point of the Control group was considered. This is surprising as animals in the Control Group were considered healthy and used for teaching. The most intriguing point was that pain scores increased at the second assessment performed 24–48 hours after the first one with no apparent reason that could influence pain scores. This was the reason the authors calculated specificity based on the first time point of the Control group. Specificity and sensitivity may be improved if the BPS

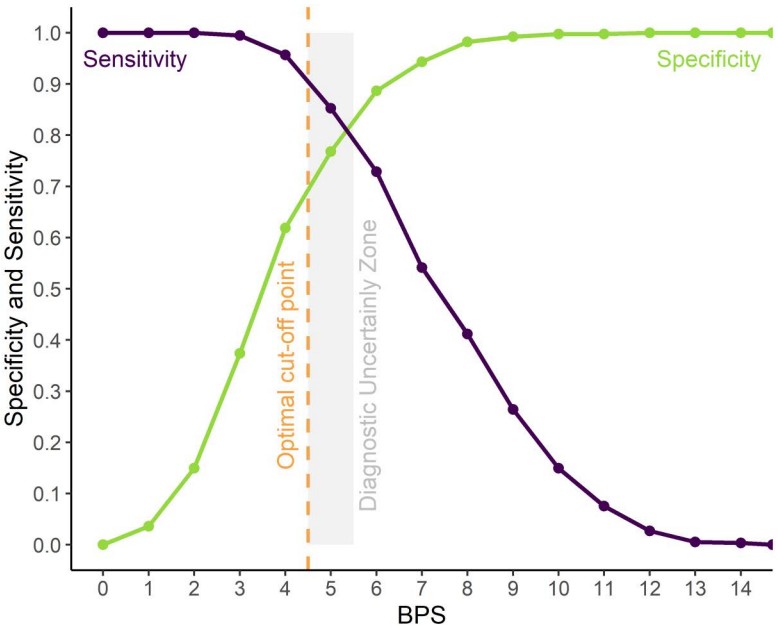

**Fig 3. Box-plot of Bovine Pain Scale (BPS) total score over time-points of perioperative period of cattle undergoing surgery (n =25; Pain Group) and healthy cows (n=11; Control Group) for all raters.** Time-points: P1, preoperatively, immediately prior to administration of sedatives; P2, 2 to 6 hours after the end of surgery; P3, 1 hour after the administration of analgesic intervention if required; P4, 24 hours after surgery; were assessed. C1, first video and assessment of animals of the control group; C2, the second video of same animal of the control group, recorded 24-48 hours after C1. The lower and upper bounds of the box respectively represent the first and third quartile of data; the horizontal line plus space inside the box indicates the median; the black diamond indicates the average of each time-point data separately; black circles indicate outlier. Different lowercase letters indicate statistical difference over the time-points (a>b); multiple comparisons were conducted by linear mixed model with post-test corrected by Bonferroni procedure ($p<0.05$).

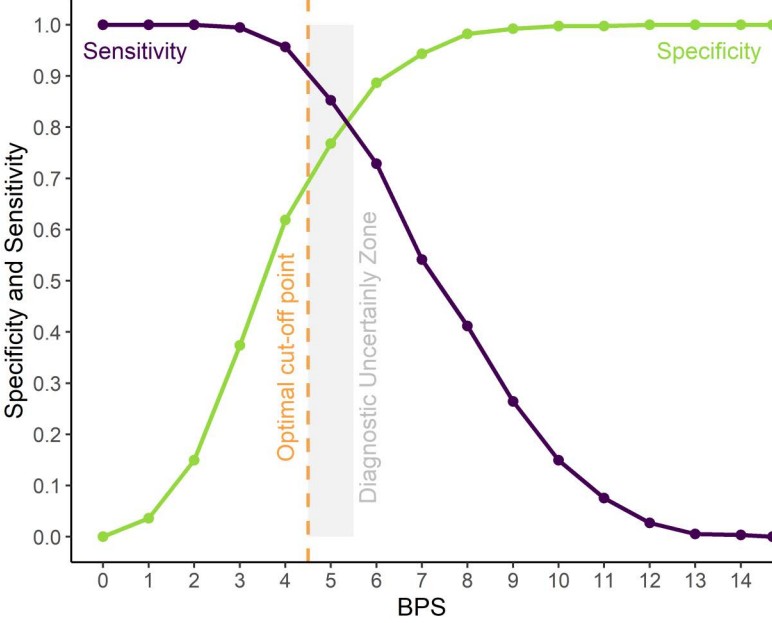

**Fig 4. ROC curve with the diagnostic uncertainty zone for the Bovine Pain Scale (BPS).** Two-graph ROC curve, CI of 1,001 replications was used to estimate the diagnostic uncertainty zone of the cut-off point of all raters, according to the Youden index [31,32]. The diagnostic uncertainty zone was 5; ≤4 indicates pain-free animals (true negatives) and ≥ 6 indicates animals suffering pain (true positives). The Youden index ≥ 5 represents the cut-off point for the indication of rescue analgesia.

is tested using a more controlled study design involving animals undergoing similar surgical procedures and standardized anesthetic and analgesic regimens. A recent study using a forest algorithm refined a pain scale and found the best pain behaviors set [43] this methodology should be used in the future to refine the Bovine Pain Scale.

The BPS detected changes in scores over time but not analgesia. We hypothesized that responsiveness after treatment was not detected due to inappropriate analgesic treatment or dosage regimens for the type and degree of pain. Pain management is challenging in cattle as approved analgesic drugs are limited due to regulatory framework related to food safety. For example, the administration of intravenous butorphanol at 0.025–0.05 mg/kg may not produce satisfactory analgesia after orthopaedic procedures and there is a concern that other opioid analgesics may produce ileus and colic in cattle, particularly when animals are to be discharged from the hospital [44–46]. Additionally, dosage regimens of analgesic drugs have not been determined in cattle. Indeed, response to rescue analgesia was not observed either for UCAPS or CPS in *Bos taurus* (Nelore) and *Bos indicus* (Angus) bulls submitted to thermal warming and orchiectomy [16], even after administration of morphine. On the other hand, the UCAPS showed responsiveness to the administration of ketoprofen and morphine in cattle undergoing orchiectomy [15].

In contrast to the inadequate results of specificity and sensitivity of BPS, its AUC of 0.90 represents a high discriminatory ability to differentiate painful and non-painful individuals. The optimum cut-off point, according to the ROC curve analysis [47], guides clinical decisions for administering analgesia in hospitalized cattle undergoing surgery. The use of cut-off scores may help when pain is unrecognized or neglected, and analgesia is required.

This study has limitations. As discussed before, the patient population was heterogeneous in intensity and type of pain, leading to poor sensitivity, item-total correlation, and internal consistency of the BPS. However, by merging two previously validated pain scales, the study offers a promising pain scoring system for cattle undergoing surgery with appropriate reliability and discriminatory ability that can be refined in the future. Previous pain assessment instruments did not address this gap of knowledge as none of them were applied to animals that had already preoperative painful conditions [14–16,36,37]. Surgical wounds may have been observed during video assessment. Future studies should perform wound bandage/cover/dressing of a similar nature in controls or sham animals in an attempt to blind observers during pain assessment. In addition, time of the day was not recorded during the assessments and video recordings, future studies should address this issue, and test if circadian rhythm can affect pain scores. This study only employed female raters during video assessment. Female veterinarians provide higher pain scores than male individuals most likely demonstrating a higher level of empathy in some studies [48]. Although female veterinarians assumed that small animals face more intense postoperative pain [49] and their pain scores are higher than male veterinarians [50–52], these results are conflicting, as higher scores were observed with male than female individuals in a study in cats [53] or when gender did not affect pain scores [54]. Finally, the majority of animals were female and of dairy breeds, hence future studies should test the BPS in males and beef cattle breeds. Future studies using artificial intelligence could be used to to rank the importance of BPS behaviors like performed for UCAPS [55].

## Conclusions

The BPS is an instrument with potential to be used for acute pain assessment in the perioperative setting in hospitalized cattle. It showed repeatability, reproducibility, high discriminatory ability and may guide the administration of analgesia. Future studies may address some limitations of the present methodology, by including a greater number of cattle of a more homogenous population with similar types of pain (soft tissue or orthopaedic procedures) and using both male and female raters. Animal welfare is a concern in farm animals and the BPS may contribute to better means of pain management in cattle undergoing surgery.

## Supporting information

**S1 Fig. Box-plot of the total sum of the Bovine Pain Scale (BPS) for 'phases' of perioperative period of cattle undergoing surgery (n = 25; Pain Group) and healthy cows (n = 11; Control Group).**
(TIF)

**S2 Fig. Smooth line of total sum of the Bovine Pain Scale (BPS) for 'raters' of perioperative period of cattle undergoing surgery (n = 25; Pain Group).** Time-points: P1, preoperatively, immediately prior to administration of sedatives; P2, 2–6 hours after the end of surgery; P3, 1 hour after the administration of analgesic intervention if required; P4, 24 hours after surgery; were assessed.
(TIF)

**S1 Table. Anaesthetic protocols of animals included in the Pain Group (n = 25).**
(DOCX)

**S1 Supplementary material. Data of Bovine Pain Scale of cattle undergoing surgery (n = 25; Pain Group) and healthy cows (n = 11; Control Group).**
(XLSX)

**S2 Supplementary material. Content Validity Index of Bovine Pain Scale.**
(XLSX)

**S3 Supplementary material. Bovine Pain Scale Training Manual.**
(DOCX)

**S4 Supplementary material. Survey Monkey Questionnaire of Bovine Pain Scale.**
(PDF)

**S5 Supplementary material. Bovine Pain Scale.**
(DOCX)

## Acknowledgments

To Dr. Alice de Oliveira, Dr. Flavia Augusta de Oliveira, Dr. Giorgia Della Rocca, Dr. Jose Ricardo Barboza Silva, and Dr. Marianne Villettaz Robichad for participating in the content validity index.

## Author contributions

**Conceptualization:** Rubia Mitalli Tomacheuski, Stelio Pacca Loureiro Luna, Paulo Vinicius Steagall.

**Data curation:** Rubia Mitalli Tomacheuski.

**Formal analysis:** Rubia Mitalli Tomacheuski, Beatriz Paglerani Monteiro, Stelio Pacca Loureiro Luna, Paulo Vinicius Steagall.

**Funding acquisition:** Rubia Mitalli Tomacheuski, Stelio Pacca Loureiro Luna, Paulo Vinicius Steagall.

**Investigation:** Rubia Mitalli Tomacheuski, Cassandra Klostermann, Diane Frank, Marilda Onghero Taffarel, Renata Haddad Pinho, Beatriz Paglerani Monteiro, André Desrochers, Sylvain Nichols, Karina Gleerup, Paulo Vinicius Steagall.

**Methodology:** Rubia Mitalli Tomacheuski, Paulo Vinicius Steagall.

**Project administration:** Rubia Mitalli Tomacheuski.

**Resources:** Stelio Pacca Loureiro Luna, Paulo Vinicius Steagall.

**Supervision:** Stelio Pacca Loureiro Luna, Paulo Vinicius Steagall.

**Validation:** Rubia Mitalli Tomacheuski, Beatriz Paglerani Monteiro, Pedro Henrique Esteves Trindade, Stelio Pacca Loureiro Luna, Paulo Vinicius Steagall.

**Visualization:** Rubia Mitalli Tomacheuski, Beatriz Paglerani Monteiro, Stelio Pacca Loureiro Luna, Paulo Vinicius Steagall.

**Writing – original draft:** Rubia Mitalli Tomacheuski, Beatriz Paglerani Monteiro.

**Writing – review & editing:** Rubia Mitalli Tomacheuski, Pedro Henrique Esteves Trindade, Stelio Pacca Loureiro Luna, Paulo Vinicius Steagall.

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
