## [Decision Letter · Decision Letter 0]

15 Jan 2025

PONE-D-24-42098Bovine Pain Scale: a novel tool for pain assessment in cattle undergoing surgery in the hospital settingPLOS ONE

Dear Dr. Tomacheuski,

Thank you for submitting your manuscript to PLOS ONE. After careful consideration, we feel that it has merit but does not fully meet PLOS ONE’s publication criteria as it currently stands. Therefore, we invite you to submit a revised version of the manuscript that addresses the points raised during the review process.

Considering the reviewers' suggestions, I ask that you make the necessary adjustments so that we can continue with the publication work.

Att

We look forward to receiving your revised manuscript.

Kind regards,

Julio Cesar de Souza, Ph.D.

Academic Editor

PLOS ONE

Journal Requirements:

2. Please include a complete copy of PLOS’ questionnaire on inclusivity in global research in your revised manuscript. Our policy for research in this area aims to improve transparency in the reporting of research performed outside of researchers’ own country or community. The policy applies to researchers who have travelled to a different country to conduct research, research with Indigenous populations or their lands, and research on cultural artefacts. The questionnaire can also be requested at the journal’s discretion for any other submissions, even if these conditions are not met.  Please find more information on the policy and a link to download a blank copy of the questionnaire here: https://journals.plos.org/plosone/s/best-practices-in-research-reporting. Please upload a completed version of your questionnaire as Supporting Information when you resubmit your manuscript

Additional Editor Comments (if provided):

Considering the reviewers' suggestions, I ask that you make the necessary adjustments so that we can continue with the publication work.

Att

Reviewers' comments:

Reviewer's Responses to Questions

**Comments to the Author**

1. Is the manuscript technically sound, and do the data support the conclusions?

Reviewer #1: Partly

Reviewer #2: Yes

Reviewer #3: Yes

2. Has the statistical analysis been performed appropriately and rigorously? 

Reviewer #1: Yes

Reviewer #2: Yes

Reviewer #3: Yes

3. Have the authors made all data underlying the findings in their manuscript fully available?

Reviewer #1: Yes

Reviewer #2: Yes

Reviewer #3: Yes

4. Is the manuscript presented in an intelligible fashion and written in standard English?

Reviewer #1: Yes

Reviewer #2: Yes

Reviewer #3: Yes

5. Review Comments to the Author

Reviewer #1: The manuscript describes the evaluation of a bovine pain measurement system which was developed by including the experience with already published systems. The aim was the discrimination of painful and non-painful individuals. 25 animals were included in the pain group and 11 animals in the control group.

The practical outcome of the study and its consequences are hard to evaluate due to two main points:

1. The results of the BPS study for both the pain group and the control group are only summarized in Supplemental Fig. S1 (a circle stands for one evaluation by one of the four raters?). This is an essential information to evaluate the study and, therefore, should not be classified as supplemental result:

- Please provide the standard deviation of the high variation within the groups for all time points.

- What is the biological relevance of the difference between an average total score of 3-5 of 18 points in the control group and 5-6 of 18 points in the pain group (and by considering the high variation within the groups) for an individual which is now subsequently tested with the BPS?

- How many of the animals of the pain group underwent analgetic treatment? In my opinion, the pain group has to be divided in two subgroups with/without analgetic treatment at least for the subsequent time points (or better for all time points?) for the statistical analyses of the results.

- Generally, it is not clear why all animals of the pain group are used together for the statistical analyses of the results. Presumably this may lead to a result which is highly specific for the study described. Does it make sense to divide the pain group according to the total score to carry out at least some of the statistical analyses for the new subgroups separately?

2. For readers not involved in the use of pain scales in cattle, some essential information is not given in the manuscript and/or by only referring to other publications, e.g.:

- What is the rationale to use the chosen nine items of the BPS and, in addition, to use them with equal classification in view of the points given?

- Results section, Tables 4 and 5: VAS: describe methodology and calculation.

- Figures: Figure legends are lacking/incomplete.

- As much abbreviations are used, please provide a list of ALL abbreviations.

The aim of the study is to improve the recognition of pain in cattle. The Discussion section gives some hints for the comparison of the study results to already published results achieved with other pain scale tools. A more systematic comparison e.g. by including a table for this comparison would enhance the information content of the manuscript. A major point in this comparison is the description of the feasibility of the different scales.

Reviewer #2: My suggestion based on PLOS ONE Standard:

Weaknesses and Suggestions

1. Internal Consistency (Lines 218–227): The internal consistency score is below the acceptable threshold. Consider discussing how this may affect clinical applications and whether modifications are planned to improve this metric.

2. Potential Bias (Lines 152–160): Surgical wounds were visible during video assessments, potentially biasing rater scores. Discuss strategies to minimize this issue in future studies.

3. Sensitivity Issues (Lines 348–354): The scale lacks sensitivity for certain pain indicators. Provide recommendations for refining these items to improve practical utility.

4. Comparison with Existing Tools (Lines 83–94): Expand comparisons with existing pain scales like UNESP-Botucatu Cattle Pain Scale (UCAPS) to emphasize the additional value of BPS.

5. Simplification of Statistical Details (Lines 182–200): Simplify complex explanations of statistical methods to enhance readability for a broader audience.

My recommendation:

Pending minor revisions addressing internal consistency, sensitivity, and clarity, the manuscript is recommended for publication in PLOS ONE.

Reviewer #3: Generally, it is a good study that enhances welfare management in animals. However, it should be clearly explained what has been done, and further clarification is needed. Some explanations are unclear as to whether they have been conducted or not. The second concern is that it is not exactly clear which study design has been used. In some parts, it appears experimental (randomized controlled trial), but it also seems to be a prospective cohort study. How compounding factors, such as environmental factors, were handled is not explained. This should be clearly addressed in the study design section. For a prospective study, the sample size is very small. How was the sample size calculated? The selection of control and exposed groups should also be clearly explained. In the results section, each output should be explained in detail; simply presenting it in table form is not enough. I observed wide gaps between raters. Find detailed comments in the attachment

6. PLOS authors have the option to publish the peer review history of their article (what does this mean? ). If published, this will include your full peer review and any attached files.

**Do you want your identity to be public for this peer review?** For information about this choice, including consent withdrawal, please see our Privacy Policy .

Reviewer #1: No

Reviewer #2: **Yes: ** Dr. Shewatatek Melaku Asefa

Reviewer #3: **Yes: ** Guash Abay Assefa

---

## [Author Response · Author response to Decision Letter 1]

4 Apr 2025

Additional Editor Comments (if provided):

Considering the reviewers' suggestions, I ask that you make the necessary adjustments so that we can continue with the publication work.

Att

Response: The authors appreciate the Reviewer´s time and effort spent reviewing this manuscript. Thank you for considering our manuscript for publication.

Reviewer #1: The manuscript describes the evaluation of a bovine pain measurement system which was developed by including the experience with already published systems. The aim was the discrimination of painful and non-painful individuals. 25 animals were included in the pain group and 11 animals in the control group.

Response: Thank you for your comments and review!

The practical outcome of the study and its consequences are hard to evaluate due to two main points:

1. The results of the BPS study for both the pain group and the control group are only summarized in Supplemental Fig. S1 (a circle stands for one evaluation by one of the four raters?). This is an essential information to evaluate the study and, therefore, should not be classified as supplemental result:

Response: This figure was added to the manuscript – figure 3. Line: 294.

- Please provide the standard deviation of the high variation within the groups for all time points.

Response: This analysis was not added because those variables are nonparametric, thus presenting it as median, first and third quartile is adequate. This was also done in other studies (Werneck et al. 2024, Luna 2020, Silva 2020).

- What is the biological relevance of the difference between an average total score of 3-5 of 18 points in the control group and 5-6 of 18 points in the pain group (and by considering the high variation within the groups) for an individual which is now subsequently tested with the BPS?

Response: A clinical study involving the development and validation of a pain scoring system does not precisely address the biological relevance of pain scores, especially when the study design does not differentiate mild, moderate and severe pain. However, the receiver operating characteristic (ROC) curve showed that scores of ≥ 5 of 18 better discriminates painful vs non-painful individuals, which true negatives and positives scoring lower or equal to 4 or higher or equal to 6, respectively. This was a clinical study, performed in a hospital environment. Clinical judgement must be used during pain assessment even using pain scoring tools. Our results showed that pain scores between 4-6 could fall into the diagnostic uncertainty zone. In this case, veterinarians should be aware that clinical judgment must be applied; animals may be reassessed as needed when presenting these scores. For example, analgesia may be administered if pain scores are 4 and the clinician feels that pain should be addressed.

- How many of the animals of the pain group underwent analgetic treatment? In my opinion, the pain group has to be divided in two subgroups with/without analgetic treatment at least for the subsequent time points (or better for all time points?) for the statistical analyses of the results.

Response: All animals received analgesic treatment (with NSAIDs or opioids) either pre-operatively or postoperatively. We did not have any animals without analgesic treatment; hence, unfortunately division into two subgroups is not possible and this procedure would not affect scale validity according to the statistical methods described in the manuscript.

- Generally, it is not clear why all animals of the pain group are used together for the statistical analyses of the results. Presumably this may lead to a result which is highly specific for the study described. Does it make sense to divide the pain group according to the total score to carry out at least some of the statistical analyses for the new subgroups separately?

Response: Please see our comment above. For the sake of scale validation regarding criterion validity, specificity and sensitivity, construct validity and responsiveness, pain scores should be pooled together and compared with controls and/or before the administration of analgesics using different types of pain and anesthetic/analgesics protocols. This approach mimics the clinical setting. Further studies may refine the Bovine Pain Scale by splitting groups according to type of surgery and drug protocols.

2. For readers not involved in the use of pain scales in cattle, some essential information is not given in the manuscript and/or by only referring to other publications, e.g.:

- What is the rationale to use the chosen nine items of the BPS and, in addition, to use them with equal classification in view of the points given?

Response: Pain produces several behavioral changes as represented in a composite pain scale. Equal weighting and associations of pain behaviors/items were tested using content validity, internal consistency, item-to-item correlation and multiple association with exploratory factor analysis. Therefore, each item of the scale has been tested to be included in the final version. Lines 124 to 132 explain the Bovine Pain Scale, each item is important for the assessment, scores 0 mean normal behavior and no pain, scores 1 and 2 represent pain related behaviors. Added to the text is an explanation. Lines: 127-128. Thanks for pointing this out. Our intention is in a future study to use artificial intelligence to weigh the importance of each behavior like performed for the Unesp-Botucatu Cattle Pain Scale (Trindade et al. 2024 - Ranking bovine pain-related behaviors using a logistic regression algorithm). This was included in discussion (lines: 465-466).

- Results section, Tables 4 and 5: VAS: describe methodology and calculation.

Response: Visual analog scale (VAS; 0-100 mm, where 1 = no pain and 100 = the worst pain imaginable) – added to both legends. This is described in Table 2 as follows: “For the sum of the BPS and VAS, the intraclass correlation coefficient (ICC; “icc{irr}”) two-way random effects model, type consistency multiple raters/measurements and its 95% CI based in 1,001 replications by the bootstrap method (“boot.ci{boot}”) was used. Interpretation of kw and ICC: very good 0.81 - 1.0; good 0.61 - 0.80; moderate 0.41 - 0.60; reasonable 0.21 - 0.4 and poor < 0.2 [21–23].”

- Figures: Figure legends are lacking/incomplete.

Response: They have been amended.

- As much abbreviations are used, please provide a list of ALL abbreviations.

Response: A list of abbreviations has been added as supplementary material. The abbreviations were described in full when they first appeared in the text.

The aim of the study is to improve the recognition of pain in cattle. The Discussion section gives some hints for the comparison of the study results to already published results achieved with other pain scale tools. A more systematic comparison e.g. by including a table for this comparison would enhance the information content of the manuscript. A major point in this comparison is the description of the feasibility of the different scales.

Response: More comparisons between the proposed scale versus UCAPS and CPS were included in Discussion (please see lines 346 and 362) and other comparisons had already been included (lines 371-373, 376-416). We appreciate the suggestion, however, the authors believe that the comparisons performed in the discussion are sufficient. The feasibility was not assessed with CPS or UCAPS. Hence, comparisons are not possible

Thank you,

Reviewer #2: My suggestion based on PLOS ONE Standard:

Weaknesses and Suggestions

1. Internal Consistency (Lines 218–227): The internal consistency score is below the acceptable threshold. Consider discussing how this may affect clinical applications and whether modifications are planned to improve this metric.

Response: Internal consistency was below the acceptable threshold due to a heterogeneous population undergoing different procedures and anesthetic protocols. We have provided additional information in the discussion (starting line 413).

2. Potential Bias (Lines 152–160): Surgical wounds were visible during video assessments, potentially biasing rater scores. Discuss strategies to minimize this issue in future studies.

Response: Strategies to address this potential bias have now been added.

3. Sensitivity Issues (Lines 348–354): The scale lacks sensitivity for certain pain indicators. Provide recommendations for refining these items to improve practical utility.

Response: This has been added to the discussion (lines 422-423). A recent study using a forest algorithm refined a pain scale and found the best pain behaviors set (Pivato et al 2025), this methodology should be used in the future to refine the bovine pain scale (lines 423-425).

4. Comparison with Existing Tools (Lines 83–94): Expand comparisons with existing pain scales like UNESP-Botucatu Cattle Pain Scale (UCAPS) to emphasize the additional value of BPS.

Response: Additional information has been added. However, comparisons among existing scales and the BPS are mostly provided in the discussion.

5. Simplification of Statistical Details (Lines 182–200): Simplify complex explanations of statistical methods to enhance readability for a broader audience.

Response: The authors followed the COSMIN guidelines (reference), which recommend statistical analyses to be provided in detail so the measurement properties of the instrument can be fully assessed. We provided a Table to facilitate reading and understanding. According to two systematic reviews, manuscripts on the developments of pain scoring instruments often have insufficient description of statistical analyses (Evangelista et al 2020, Tomacheuski et al 2023).

My recommendation:

Pending minor revisions addressing internal consistency, sensitivity, and clarity, the manuscript is recommended for publication in PLOS ONE.

Response: Thank you very much for your comments and review!

My suggestion based on PLOS ONE Standard:

1. Research Originality: The study presents a new method for measuring pain in hospitalized cattle: the Bovine Pain Scale (BPS). This uniqueness may be seen throughout the work, but it is particularly highlighted in the Introduction (lines 67–94) and Abstract (lines 39–62).

Response: Thank you very much for your comments and review!

2. Novelty of Results: The results seem to be unique and haven't been published before, especially the scale's validation metrics (lines 202–227). Though there isn't a comparable all-inclusive tool, think about making it clear in the introduction.

Response: This has been added in the introduction. Lines: 91-93

3. Technical and Analytical Standards: Materials and Methods (lines 182–200) provide specifics on the statistical techniques, such as ROC curve analysis and intra-class correlation coefficients. There are problems with internal consistency in Results (lines 218–227), when scores fall below the permissible cutoff of 0.6. This calls for explanation or discussion of the consequences for implications for clinical applications.

Response: We have now addressed the issue of internal consistency in the discussion (starting line 413).

4. Data-Driven Conclusions: The Discussion (lines 316–354) provides an effective summary of the conclusions, which are backed up by data. As may be observed in lines 344–348, the authors admit limits in construct validity and sensitivity.

Response: Thank you

5. Language and Clarity: The manuscript is generally well-written and clear. To facilitate wider accessibility, some sections, including the explanations of statistical analysis (lines 182–200), can benefit from simplification.

Response: The authors followed the COSMIN guidelines (reference), which recommend statistical analyses to be provided in detail so the measurement properties of the instrument can be fully assessed. We provided a Table to facilitate reading and understanding. According to two systematic reviews, manuscripts on the developments of pain scoring instruments often have insufficient description of statistical analyses (Evangelista et al 2020, Tomacheuski et al 2023).

6. Ethical Standards: The Materials and Methods (lines 96–116) contain extensive documentation of ethical compliance, including the Institutional Animal Care and Use Committee's approval and measures to reduce animal suffering.

Response: Thank you

7. Reporting criteria and Data Availability: The validation procedure clearly complies with COSMIN (Consensus-based Standards for the selection of health Measurement Instruments) criteria (lines 97–102), and the Additional Information section makes clear that data is available.

Response: Thank you

Strengths:

The manuscript addresses a critical gap in cattle pain management (lines 67–94).

High inter- and intra-rater reliability (lines 210–216) demonstrates the scale's robustness.

Strong correlation with visual analog scales (VAS) supports criterion validity (line 353).

Reviewer #3: Generally, it is a good study that enhances welfare management in animals. However, it should be clearly explained what has been done, and further clarification is needed. Some explanations are unclear as to whether they have been conducted or not. The second concern is that it is not exactly clear which study design has been used. In some parts, it appears experimental (randomized controlled trial), but it also seems to be a prospective cohort study. How compounding factors, such as environmental factors, were handled is not explained. This should be clearly addressed in the study design section. For a prospective study, the sample size is very small. How was the sample size calculated? The selection of control and exposed groups should also be clearly explained. In the results section, each output should be explained in detail; simply presenting it in table form is not enough. I observed wide gaps between raters. Find detailed comments in the attachment

Response: Thank you very much for your comments and review!

Comments

Generally, it is a good study that enhances welfare management in animals. However, it should be clearly explained what has been done, and further clarification is needed. Some explanations are unclear as to whether they have been conducted or not. The second concern is that it is not exactly clear which study design has been used. In some parts, it appears experimental (randomized controlled trial), but it also seems to be a prospective cohort study. How compounding factors, such as environmental factors, were handled is not explained. This should be clearly addressed in the study design section. For a prospective study, the sample size is very small. How was the sample size calculated? The selection of control and exposed groups should also be clearly explained. In the results section, each output should be explained in detail; simply presenting it in table form is not enough. I observed wide gaps between raters. My detailed comments are given below.

Response: The study design was a prospective, observational clinical study as described in the first two lines of the M&M. A detailed description of the animal population has been described in the Methods. The Pain Group consisted of patients recruited at the CHUV (veterinary teaching hospital) with appropriate inclusion and exclusion criteria. The sample size was based on a previous study (Tomacheuski et al 2023) (lines 201-204), but the small sample size was due to the impact of COVID 19 (this is addressed in the manuscript). This study was performed at the CHUV and environment (single stalls) is described in the Methods with appropriate acclimation protocols to cameras and personnel, etc. The Control Group consisted of animals from the teaching colony that were deemed healthy based on history, physical examination and regular blood work. The table demonstrates the demographics of the pain group. The available videos can also provide additional views of the hospital facilities and how animals were assessed for pain.

1. Line 45-46: Videos were recorded before, 2 to 6 hours after surgery, 1 hour after the administration of analgesia and 24 hours after surgery. How long before the surgery were the videos taken?

Response: It is addressed in the lines 136-137 “P1, preoperatively, immediately prior to adminis

---

## [Editor Report · Decision Letter 1]

13 Apr 2025

Bovine Pain Scale: a novel tool for pain assessment in cattle undergoing surgery in the hospital setting

PONE-D-24-42098R1

Dear Dr. Tomacheuski,

We’re pleased to inform you that your manuscript has been judged scientifically suitable for publication and will be formally accepted for publication once it meets all outstanding technical requirements.

Kind regards,

Julio Cesar de Souza, Ph.D.

Academic Editor

PLOS ONE

Additional Editor Comments (optional):

Considering the reviewers' suggestions and that the authors had the opportunity to make adjustments, as well as justifying what they thought should remain as is, I am in favor of publishing the paper.
---

## [Editor Report · Acceptance letter]

PONE-D-24-42098R1

PLOS ONE

Dear Dr. Tomacheuski,

I'm pleased to inform you that your manuscript has been deemed suitable for publication in PLOS ONE. Congratulations! Your manuscript is now being handed over to our production team.

Kind regards,

on behalf of

Dr. Julio Cesar de Souza

Academic Editor

PLOS ONE